# Egg and cholesterol consumption and mortality from cardiovascular and different causes in the United States: A population-based cohort study

**Pan Zhuang**[1,2,4], **Fei Wu**[3], **Lei Mao**[3], **Fanghuan Zhu**[1,2,4], **Yiju Zhang**[1,2], **Xiaoqian Chen**[1,2,4], **Jingjing Jiao**[3]*, **Yu Zhang**[1,2,4]*

1 National Engineering Laboratory of Intelligent Food Technology and Equipment, Fuli Institute of Food Science, Zhejiang University, Hangzhou, Zhejiang, China, 2 Key Laboratory for Agro-Products Postharvest Handling of Ministry of Agriculture and Rural Affairs, Zhejiang Key Laboratory for Agro-Food Processing, Zhejiang University, Hangzhou, Zhejiang, China, 3 Department of Nutrition, School of Public Health, Department of Nutrition of Affiliated Second Hospital, Zhejiang University School of Medicine, Hangzhou, Zhejiang, China, 4 Department of Food Science and Nutrition, College of Biosystems Engineering and Food Science, Zhejiang University, Hangzhou, Zhejiang, China

* jingjingjiao@zju.edu.cn (JJ); y_zhang@zju.edu.cn (YZ)

**Data Availability Statement:** Data cannot be shared publicly because of the sensitive nature of the data collected for this study. Data are available from the US National Cancer Institute Division of Cancer Epidemiology & Genetics (contact via Linda M. Liao, e-mail: liaolm@mail.nih.gov) for

## Abstract

### Background

Whether consumption of egg and cholesterol is detrimental to cardiovascular health and longevity is highly debated. Data from large-scale cohort studies are scarce. This study aimed to examine the associations of egg and cholesterol intakes with mortality from all causes, cardiovascular disease (CVD), and other causes in a US population.

### Methods and findings

Overall, 521,120 participants (aged 50–71 years, mean age = 62.2 years, 41.2% women, and 91.8% non-Hispanic white) were recruited from 6 states and 2 additional cities in the US between 1995 and 1996 and prospectively followed up until the end of 2011. Intakes of whole eggs, egg whites/substitutes, and cholesterol were assessed by a validated food frequency questionnaire. Cause-specific hazard models considering competing risks were used, with the lowest quintile of energy-adjusted intake (per 2,000 kcal per day) as the reference. There were 129,328 deaths including 38,747 deaths from CVD during a median follow-up of 16 years. Whole egg and cholesterol intakes were both positively associated with all-cause, CVD, and cancer mortality. In multivariable-adjusted models, the hazard ratios (95% confidence intervals) associated with each intake of an additional half of a whole egg per day were 1.07 (1.06–1.08) for all-cause mortality, 1.07 (1.06–1.09) for CVD mortality, and 1.07 (1.06–1.09) for cancer mortality. Each intake of an additional 300 mg of dietary cholesterol per day was associated with 19%, 16%, and 24% higher all-cause, CVD, and cancer mortality, respectively. Mediation models estimated that cholesterol intake contributed to 63.2% (95% CI 49.6%–75.0%), 62.3% (95% CI 39.5%–80.7%), and 49.6% (95% CI

researchers who meet the criteria for access to confidential data.

**Funding:** Y.Z. is supported by the National Key Research and Development Program of China (2017YFC1600500). The funders had no role in study design, data collection and analysis, decision to publish, or preparation of the manuscript.

**Competing interests:** The authors have declared that no competing interests exist.

**Abbreviations:** AD, Alzheimer disease; BMI, body mass index; CI, confidence interval; CVD, cardiovascular disease; FFQ, food frequency questionnaire; HDL-c, high-density lipoprotein cholesterol; HEI-2015, Healthy Eating Index–2015; HR, hazard ratio; LDL-c, low-density lipoprotein cholesterol; RD, respiratory disease.

31.9%–67.4%) of all-cause, CVD, and cancer mortality associated with whole egg consumption, respectively. Egg white/substitute consumers had lower all-cause mortality and mortality from stroke, cancer, respiratory disease, and Alzheimer disease compared with non-consumers. Hypothetically, replacing half a whole egg with equivalent amounts of egg whites/substitutes, poultry, fish, dairy products, or nuts/legumes was related to lower all-cause, CVD, cancer, and respiratory disease mortality. Study limitations include its observational nature, reliance on participant self-report, and residual confounding despite extensive adjustment for acknowledged dietary and lifestyle risk factors.

## Conclusions

In this study, intakes of eggs and cholesterol were associated with higher all-cause, CVD, and cancer mortality. The increased mortality associated with egg consumption was largely influenced by cholesterol intake. Our findings suggest limiting cholesterol intake and replacing whole eggs with egg whites/substitutes or other alternative protein sources for facilitating cardiovascular health and long-term survival.

## Trial registration

ClinicalTrials.gov NCT00340015.

## Author summary

### Why was this study done?

- The 2015–2020 dietary guidelines for Americans recommend eating as little dietary cholesterol as possible while consuming a healthy eating pattern, but also state cholesterol is not a nutrient of concern for overconsumption.

- There is limited and inconclusive evidence on the associations of egg or dietary cholesterol intake with all-cause and cardiovascular disease mortality.

- Associations of egg or dietary cholesterol intake with mortality from non-cardiovascular causes, including cancer, respiratory disease, diabetes, and Alzheimer disease, are largely unknown.

### What did the researchers do and find?

- We used data on 521,120 participants from the NIH-AARP Diet and Health Study with a median follow-up of 16 years to assess the associations of egg and cholesterol intakes with all-cause and cause-specific mortality.

- Whole egg and cholesterol intakes were positively associated with risk of all-cause and major cause-specific mortality, whereas egg white/substitute consumption was inversely associated with mortality.

- Mediation models estimated that the increased mortality associated with whole egg intake was largely influenced by cholesterol intake.

- Replacing half a whole egg with egg whites/substitutes or other protein sources containing less cholesterol is associated with lower overall mortality and mortality from major causes, including cardiovascular disease, cancer, and respiratory disease.

### What do these findings mean?

- The current recommendations for egg and dietary cholesterol intake from the US dietary guidelines might lead to increases in cholesterol intake, which could be detrimental to the prevention of premature death.

- Clinicians and policy makers should continue to highlight limiting cholesterol intake in the US dietary recommendations, considering our results.

- The US dietary guidelines may recommend replacing whole eggs with egg whites/substitutes or alternative protein sources for facilitating cardiovascular health and long-term survival.

## Introduction

Eggs are a good source of high-quality protein, vitamins, and other bioactive nutrients such as lecithins and carotenoids [1], and are recommended as part of a healthy diet in the 2015–2020 US dietary guidelines [2]. However, many people do not consume eggs or egg yolks due to the high amount of cholesterol (approximately 186 mg cholesterol/egg), which might be a dietary risk factor for cardiovascular health [3]. Recently, such public concern has been intensified by a pooled study of 6 prospective US cohorts that linked egg and cholesterol consumption to higher risk of cardiovascular disease (CVD), CVD mortality, and all-cause mortality [4]. In the past decades, limiting dietary cholesterol intake to 300 mg/day was initially recommended for CVD prevention [5]. However, the most recent meta-analysis showed heterogeneous results about the relationship between dietary cholesterol and the risk of CVD death, coronary artery disease, and stroke [6]. Owing to the inadequate evidence, the American Heart Association and American College of Cardiology [7,8] and the 2015–2020 US dietary guidelines [2,9] have not carried forward the upper limit for dietary cholesterol. In addition, cholesterol often coexists with saturated fat and animal protein in foods [2]. The independent effect of cholesterol and the interaction between cholesterol and these nutrients on overall health have yet to be elucidated. With regard to eggs, previous observational studies reported inconsistent results, and most meta-analyses concluded there were not significant associations with CVD or all-cause mortality among the general population [10–14]. However, the studies included in these meta-analyses had relatively small sample sizes or few cases, which may not support robust effect estimates when examining the associations with mortality from various causes. Furthermore, whether the associations differ between whole eggs and egg whites/substitutes, and whether the associations are modified by cooking methods such as frying, remains unclear.

To fill these knowledge gaps, we assessed the associations of egg and dietary cholesterol intake with all-cause and cause-specific mortality in the NIH-AARP Diet and Health Study, which has a large number of participants (*n* = 521,120) and death cases (*n* = 129,328). We

hypothesized that egg and cholesterol intakes would have significant associations with all-cause, CVD, and other cause-specific mortality.

## Methods

### Study population

The NIH-AARP study is a large prospective cohort initiated October 1995–May 1996 when a questionnaire was mailed to 3.5 million AARP (formerly the American Association of Retired Persons) members aged 50–71 years living in 6 US states (California, Florida, Louisiana, New Jersey, North Carolina, and Pennsylvania) and 2 additional metropolitan areas (Atlanta, Georgia, and Detroit, Michigan) [15]. The validated and comprehensive questionnaires covered a broad range of information, including demographics, lifestyle characteristics, and diet history. All participants provided written informed consent.

Among 567,169 participants who completed and returned satisfactory questionnaires at baseline, we excluded duplicate questionnaires, proxy responders, withdrawals, participants who moved out of the cancer registry ascertainment area or died before entry, individuals with 0 person-years of follow-up, and individuals with extreme daily total energy intake (<800 or >4,200 kcal/d for men and <600 or >3,500 kcal/d for women). The final analytic cohort consisted of 521,120 participants (Fig A in S1 Data). This study is reported as per the Strengthening the Reporting of Observational Studies in Epidemiology (STROBE) Statement (see S1 STROBE Checklist).

### Ethics statement

The study was approved by the Special Studies Institutional Review Board of the National Cancer Institute.

### Dietary assessment

Dietary intake was measured using a validated food frequency questionnaire (FFQ) with 124 items developed as the Diet History Questionnaire (DHQ) by the National Cancer Institute [15,16]. The FFQ inquired about the frequency of food and beverage consumption and portion sizes over the last year. The questionnaire asked the numbers of eggs consumed per day, week, month, or year. The participants were also asked to report the types of eggs (whole eggs, egg whites, or cholesterol-free commercial egg substitutes) and cooking methods for eggs (e.g., fried/sauteed in oil, poached, or boiled). The 1994–1996 USDA Continuing Survey of Food Intakes by Individuals [17] was used to convert frequency of consumption and portion size for each food item into intakes, including intakes of eggs, other foods, and dietary cholesterol. Red meat intake included all types of unprocessed (pork, beef, steak, liver, and hamburger) and processed red meat (bacon, hot dogs, cold cuts, ham, and sausage). Fish intake included all types of finfish and shellfish. Poultry intake included turkey, chicken, ground poultry, and the processed poultry components of turkey or chicken cold cuts. In validation studies, estimated correlation coefficients between true and FFQ-reported energy-adjusted intakes were higher than 0.5 for most of the food groups (0.7 in men and 0.6 in women for eggs) [16] and 0.7 for cholesterol [18]. Based on the 2015–2020 US dietary guidelines, the Healthy Eating Index–2015 (HEI-2015) score was used as previously described [19] to evaluate the adherence to an overall healthy dietary pattern.

### Covariates

Information on lifestyle characteristics and potential confounders was collected at baseline (n = 567,169), including age, sex, race, marital status, household income, education, alcohol

drinking, cigarette smoking, weight, height, physical activity, and history of hypertension, high blood cholesterol level, and chronic diseases [15]. Data on the use of cholesterol-lowering medications were collected in the resurvey (*n* = 293,918).

## Death ascertainment

All the participants were followed for address changes via the US Postal Service National Change of Address database, responses to study-related mailings, and direct notifications from cohort members. Mortality data were obtained by linkage to the National Death Index Plus maintained by the National Center for Health Statistics. We used the codes of the International Classification of Diseases 9th and 10th revisions to classify cause-specific mortality into 22 categories (Table A in S1 Data). Person-years of follow-up was calculated from the return date of the baseline questionnaire to death or the end date of follow-up (31 December 2011).

## Statistical analysis

The analysis plan was developed prospectively (S1 Text), with later adjustments, such as adding restricted-cubic-spline regression analyses and mediation analyses according to reviewers' suggestions, and no data-driven changes took place. We selected whole eggs, egg whites/substitutes, and dietary cholesterol as primary exposures in the current analysis. Energy-adjusted intakes of whole eggs, cholesterol, and other dietary factors were evaluated using the nutrient density method [20], with intakes calculated per 2,000 kcal per day and divided into quintiles for the entire study. Owing to overall low consumption, egg white/substitute consumption was analyzed as a categorical variable (non-consumers and consumers). We used cause-specific hazard models [21,22] that took competing risks into consideration to estimate hazard ratios (HRs) and 95% confidence intervals (CIs) of all-cause and cause-specific mortality according to the categories of egg and cholesterol consumption. We further investigated the proportional hazards assumption using the Kolmogorov-type supremum test and observed no evidence of violation. We then established stepwise models to adjust for the covariates of known or suspected risk factors [23,24] for mortality. Model 1 was adjusted for age and sex. Model 2 was further adjusted for body mass index (BMI), race, education, marital status, household income, cigarette smoking, alcohol drinking, vigorous physical activity, usual activity at work, and history of hypertension, hypercholesterolemia, heart disease, stroke, cancer, and diabetes at baseline. For eggs, model 3 was additionally adjusted for total energy intake and consumption of egg whites/substitutes (for whole egg analysis) or whole eggs (for egg white/substitute analysis). Based on model 3, we further adjusted for dietary cholesterol (model 4), other major food groups (model 5), or an HEI-2015 score (model 6) to determine whether the associations for egg consumption could be driven by cholesterol intake, affected by other dietary factors, or explained by the underlying dietary pattern, respectively. For dietary cholesterol, based on model 2, we additionally adjusted for total energy and intakes of cholesterol-containing foods (eggs, red meat, fish, poultry, and dairy products; model 3), non-cholesterol-containing foods (fruit, vegetables, potatoes, nuts/legumes, whole grains, refined grains, coffee, and sugar-sweetened beverages; model 4), HEI-2015 score (model 5), or nutrients correlated with cholesterol (saturated fat, polyunsaturated fat, monounsaturated fat, trans fat, animal protein, fiber, and sodium; model 6). We calculated tests for linear trend by assigning the median values to each category of intake as a continuous variable. We created indicator variables for missing data (<5%) in each covariate if necessary. We also evaluated the effects of a fixed increase in whole egg (half a whole egg, approximately 25 g) [4] and dietary cholesterol (300 mg/day) [25] intakes on mortality based on the final multivariable models, and conducted restricted-cubic-spline regression analysis to model the associations. Given that the proportion of excess

mortality associated with egg intake might be attributed to the egg-associated nutrients, including cholesterol, saturated fat, and animal protein, we further conducted mediation analysis [26] to estimate the mediation proportion with 95% CI of cholesterol, saturated fat, and animal protein intakes for whole egg consumption (quintiles) by using the publicly available "mediate" SAS macro (https://cdn1.sph.harvard.edu/wp-content/uploads/sites/271/2012/08/mediate.pdf).

We built several models to estimate the changes in mortality risk for hypothetically replacing half an egg/day with equivalent amounts of egg whites/substitutes or alternative protein sources including poultry, fish, dairy products, and nuts/legumes. The substitution models simultaneously included these protein-rich foods as continuous variables, and total energy intake, other major food groups, and non-dietary covariates. The difference between regression coefficients and in variances and covariances was used to derive the HRs and 95% CIs of the substitution analyses [27].

In secondary analyses, we further separately analyzed the associations for fried eggs and non-fried eggs according to the cooking method, and the associations for egg whites and egg substitutes. Subgroup analyses were also performed according to important potential effect modifiers, and $P$ values for interaction were computed by the likelihood ratio test. We also conducted sensitivity analyses by categorizing whole egg and cholesterol intakes by convenient consumption cutoffs, excluding participants who had extreme BMIs ($<18.5$ or $>40$ kg/m$^2$); adjusting for perceived health condition; adjusting for a propensity score [28] to further control potential residual confounding from measured variables; further adjusting for the use of vitamin supplements and aspirin; adjusting for processed meat consumption (processed red meat and processed white meat); adjusting for the use of cholesterol-lowering medications; excluding the initial 5 years of follow-up; excluding those who had CVD, cancer, or diabetes at baseline; or censoring participants at 8 years of follow-up (midpoint).

We performed statistical analyses with SAS version 9.4 (SAS Institute, Cary, NC, US) and considered 2-sided $P < 0.05$ to be statistically significant.

## Results

### Egg and cholesterol intakes and baseline characteristics

The overall median intakes of whole eggs and cholesterol were 8 g/2,000 kcal/day and 208 mg/2,000 kcal/day, respectively. At baseline, participants with higher whole egg consumption had a higher BMI and lower household income. They were less educated, less physically active, more likely to smoke and have a high cholesterol level, and less likely to take aspirin. They also had higher red meat intake; lower intakes of fruit, dairy products, and sugar-sweetened beverages; and lower HEI-2015 score (Table 1). The characteristics of participants according to cholesterol and egg white/substitute consumption are shown in Tables B and C in S1 Data, respectively. The Spearman correlations between egg or cholesterol consumption and a range of dietary factors are shown in Table D in S1 Data.

### All-cause mortality

During a mean follow-up of 16 years (7,307,097 person-years), overall 129,328 deaths occurred. Whole egg consumption was strongly associated with higher all-cause mortality in the age- and sex-adjusted model (model 1; $P$ for trend $< 0.001$). The positive relationship remained significant after further adjusting for other demographic characteristics and dietary factors (models 2, 3, 5, and 6; $P$ for trend $< 0.001$) but was not significant after further adjusting for cholesterol (model 4; $P$ for trend $= 0.64$; Table 2). In contrast, egg white/substitute consumption was significantly associated with lower all-cause mortality in all models (all $P$

**Table 1. Baseline characteristics of participants from the NIH-AARP Diet and Health Study according to whole egg consumption.**

| Characteristics | Quintiles of whole egg consumption | | | | |
|---|---|---|---|---|---|
| | Q1 | Q2 | Q3 | Q4 | Q5 |
| Range (g/2,000 kcal/day) | ≤0.9 | 1.0–5.2 | 5.3–10.3 | 10.4–19.0 | ≥19.1 |
| N | 104,224 | 104,224 | 104,224 | 104,224 | 104,224 |
| Age (years) | 62.6 (5.3) | 61.9 (5.4) | 62.0 (5.4) | 62.2 (5.3) | 62.3 (5.3) |
| Male (%) | 59.2 | 55.7 | 58.7 | 58.3 | 62.0 |
| Ethnicity (%) | | | | | |
| Non-Hispanic white | 90.7 | 91.6 | 92.4 | 92.8 | 91.4 |
| Non-Hispanic black | 4.1 | 4.0 | 3.3 | 2.9 | 3.8 |
| Other/missing | 5.2 | 4.4 | 4.2 | 4.3 | 4.8 |
| BMI (kg/m$^2$) | 26.4 (4.8) | 26.7 (4.9) | 27.1 (5.0) | 27.2 (5.0) | 28.0 (5.5) |
| Married (%) | 68.8 | 66.8 | 68.9 | 70.4 | 67.1 |
| College graduate or postgraduate (%) | 43.3 | 40.1 | 38.6 | 38.4 | 34.7 |
| Annual household income (US dollars)[a] | 55,111 (24,694) | 54,432 (24,125) | 54,064 (23,413) | 54,094 (23,358) | 52,163 (22,450) |
| Current smoker (%) | 6.9 | 11.0 | 12.3 | 12.6 | 15.6 |
| Physical activity ≥5 times/week (%) | 25.2 | 19.8 | 17.8 | 16.9 | 16.0 |
| Hypertension (%) | 25.7 | 22.0 | 22.6 | 23.3 | 23.5 |
| High cholesterol level (%) | 18.2 | 26.9 | 27.7 | 28.6 | 29.6 |
| Heart disease (%) | 23.2 | 11.6 | 11.7 | 11.6 | 11.9 |
| Stroke (%) | 2.7 | 1.9 | 1.9 | 1.9 | 2.2 |
| Cancer (%) | 8.9 | 8.6 | 8.9 | 9.2 | 9.4 |
| Diabetes (%) | 9.8 | 6.8 | 7.7 | 8.6 | 12.6 |
| Fair or poor health (%) | 14.0 | 11.3 | 11.5 | 12.2 | 15.5 |
| Currently using vitamin supplements (%) | 58.8 | 55.3 | 55.2 | 55.5 | 53.7 |
| Daily use of aspirin (%) | 20.6 | 13.9 | 13.5 | 13.6 | 12.9 |
| Egg white/substitute consumer (%) | 66.0 | 1.9 | 1.4 | 0.9 | 0.4 |
| Daily dietary intake[b] | | | | | |
| Total energy (kcal) | 1,723.7 (645.5) | 1,842.5 (643.4) | 1,880.7 (768.0) | 1,794.0 (644.7) | 1,763.4 (666.6) |
| Alcohol from alcoholic drinks (g) | 9.6 (23.4) | 11.9 (27.3) | 12.6 (27.9) | 11.3 (24.2) | 10.3 (21.3) |
| Whole egg (g) | 0.1 (0.2) | 3.0 (1.2) | 7.6 (1.5) | 14.0 (2.4) | 34.7 (19.8) |
| Cholesterol (mg) | 148.7 (57.1) | 175.1 (55.6) | 204.4 (53.4) | 236.8 (53.1) | 333.3 (103.5) |
| Total fat (% of energy) | 25.9 (7.5) | 29.2 (7.4) | 30.7 (7.0) | 31.6 (6.8) | 33.9 (7.2) |
| Saturated fat (% of energy) | 7.6 (2.6) | 9.1 (2.8) | 9.7 (2.7) | 10.0 (2.7) | 10.7 (2.9) |
| Monounsaturated fat (% of energy) | 9.7 (3.1) | 10.9 (3.0) | 11.6 (2.9) | 12.0 (2.8) | 12.9 (2.9) |
| Polyunsaturated fat (% of energy) | 6.5 (2.2) | 6.8 (2.3) | 7.0 (2.1) | 7.2 (2.1) | 7.5 (2.2) |
| Trans fat (% of energy) | 1.8 (0.8) | 2.0 (0.8) | 2.1 (0.8) | 2.2 (0.8) | 2.3 (0.9) |
| Animal protein (g) | 46 (19) | 45 (16) | 47 (15) | 48 (15) | 50 (16) |
| Fiber (g) | 26 (9) | 22 (8) | 21 (7) | 21 (7) | 19 (7) |
| Sodium (mg) | 3,132 (623) | 2,965 (597) | 3,005 (557) | 3,053 (532) | 3,105 (546) |
| Red meat (g) | 50 (40) | 65 (41) | 73 (41) | 76 (41) | 81 (44) |
| Poultry (g) | 55 (51) | 44 (42) | 43 (37) | 43 (37) | 40 (37) |
| Fish (g) | 25 (27) | 21 (22) | 21 (20) | 22 (20) | 22 (22) |
| Dairy products (g) | 400 (337) | 397 (337) | 382 (308) | 373 (292) | 344 (286) |
| Nuts/legumes (g) | 18 (22) | 15 (18) | 15 (16) | 14 (15) | 14 (15) |
| Fruit (g) | 507 (357) | 446 (348) | 410 (308) | 393 (282) | 362 (279) |
| Vegetables (g) | 399 (229) | 336 (196) | 324 (176) | 322 (167) | 313 (173) |
| Potatoes (g) | 59 (48) | 56 (44) | 57 (42) | 59 (41) | 59 (43) |
| Whole grains (g) | 39 (27) | 33 (25) | 31 (23) | 31 (22) | 29 (22) |

*(Continued)*

**Table 1.** (Continued)

| Characteristics | Quintiles of whole egg consumption | | | | |
|---|---|---|---|---|---|
| | Q1 | Q2 | Q3 | Q4 | Q5 |
| Refined grains (g) | 142 (50) | 133 (48) | 132 (45) | 132 (43) | 127 (44) |
| Coffee (g) | 881 (868) | 887 (865) | 938 (862) | 960 (815) | 1065 (923) |
| Sugar-sweetened beverages (g) | 696 (722) | 689 (702) | 655 (649) | 620 (601) | 638 (684) |
| Healthy Eating Index–2015 score | 71.2 (8.9) | 68.0 (9.6) | 67.4 (9.3) | 67.2 (9.2) | 65.2 (9.6) |

Values are mean (SD) or percentage unless stated otherwise.

[a]Household income in 1999.

[b]Dietary intakes were energy-adjusted using the nutrient density method to intake per 2,000 kcal per day unless stated otherwise.

BMI, body mass index; Q, quintile.

trend < 0.001; Table E in S1 Data). Egg white/substitute consumers had a 7% (HR 0.93; 95% CI 0.91–0.95; $P$ < 0.001) lower risk of all-cause mortality compared with non-consumers. Dietary cholesterol intake was positively associated with all-cause mortality in all models (all $P$ trend < 0.001; Table 3), including the model adjusted for cholesterol-containing foods (model 3), non-cholesterol-containing foods (model 4), overall dietary pattern (model 5), and other nutrients (model 6). Each intake of an additional 300 mg of dietary cholesterol/day was associated with a 19% (HR 1.19; 95% CI 1.16–1.22; $P$ < 0.001) higher risk of all-cause mortality (Fig 1; Table F in S1 Data). Each additional half a whole egg/day was associated with a 7% (HR 1.07; 95% CI 1.06–1.08; $P$ < 0.001) higher risk of all-cause mortality. Restricted-cubic-spline regression profiles also show similar results for whole egg and cholesterol intakes (Fig B in S1 Data; not included in the prespecified analysis plan).

## CVD mortality and cancer mortality

After multivariable adjustment, intakes of whole eggs and cholesterol were both positively associated with CVD mortality (both $P$ trend < 0.001), whereas the association for egg white/substitute consumption did not appear significant ($P$ trend = 0.40) (Tables 2 and 3; Table E in S1 Data). We observed similar associations with heart disease mortality (Table G in S1 Data). For stroke mortality, we found a positive association for whole egg consumption but inverse association for egg white/substitute consumption. With regard to cancer mortality, we also observed positive associations for the intakes of whole eggs and cholesterol ($P$ for trend < 0.001) and an inverse association for egg white/substitute consumption ($P$ for trend = 0.005). Each additional half a whole egg/day was associated with a 7% higher risk of CVD mortality and cancer mortality. Each intake of an additional 300 mg of dietary cholesterol/day was associated with a 16% and 24% higher risk of CVD mortality and cancer mortality, respectively (Fig 1; Table F in S1 Data). In the mediation analysis, we found that 63.2% (95% CI 49.6%–75.0%), 62.3% (95% CI 39.5%–80.7%), and 49.6% (95% CI 31.9%–67.4%) of increased all-cause, CVD, and cancer mortality associated with whole egg consumption was influenced by cholesterol intake, respectively (all $P$ < 0.001; Table H in S1 Data; not included in the prespecified analysis plan).

## Other cause-specific mortality

Whole egg and cholesterol intakes were associated with higher mortality from respiratory disease (RD) and diabetes, but with lower mortality from Alzheimer disease (AD) (Tables I and J in S1 Data). We also observed borderline positive associations between whole egg

**Table 2. Associations of whole egg consumption with all-cause, CVD, and cancer mortality.**

| Outcome and model | Quintiles of whole egg consumption | | | | | P trend | HR (95% CI) for half an egg/day |
| --- | --- | --- | --- | --- | --- | --- | --- |
| | Q1 | Q2 | Q3 | Q4 | Q5 | | |
| Median intake (IQR) (g/2,000 kcal/day) | 0.0 (0.0 to 0.0) | 3.0 (1.9 to 4.1) | 7.5 (6.3 to 8.9) | 13.7 (12.0 to 16.0) | 28.7 (23.0 to 38.8) | | |
| **Total mortality** | 1,467,932 | 1,481,515 | 1,468,962 | 1,463,029 | 1,425,659 | | |
| Number of deaths/participants | 25,343/104,224 | 23,273/104,224 | 24,921/104,224 | 25,507/104,224 | 30,284/104,224 | | |
| Person-years of follow-up | 1,467,932 | 1,481,515 | 1,468,962 | 1,463,029 | 1,425,659 | | |
| Model 1[a] | 1.00 | 0.97 (0.96 to 0.99) | 1.04 (1.02 to 1.06) | 1.05 (1.04 to 1.07) | 1.28 (1.26 to 1.30) | <0.001 | 1.17 (1.16–1.17) |
| Model 2[b] | 1.00 | 1.04 (1.02 to 1.05) | 1.08 (1.06 to 1.10) | 1.09 (1.07 to 1.11) | 1.17 (1.15 to 1.19) | <0.001 | 1.08 (1.07–1.09) |
| Model 3[c] | 1.00 | 1.04 (1.02 to 1.06) | 1.08 (1.06 to 1.10) | 1.10 (1.08 to 1.12) | 1.18 (1.16 to 1.20) | <0.001 | 1.08 (1.07–1.09) |
| Model 4[d] | 1.00 | 0.96 (0.94 to 0.98) | 0.98 (0.96 to 1.00) | 0.97 (0.95 to 1.00) | 0.98 (0.95 to 1.01) | 0.64 | 0.98 (0.97–1.00) |
| Model 5[e] | 1.00 | 1.01 (1.00 to 1.03) | 1.05 (1.03 to 1.07) | 1.07 (1.05 to 1.09) | 1.14 (1.12 to 1.16) | <0.001 | 1.07 (1.07–1.08) |
| Model 6[f] | 1.00 | 1.02 (1.01 to 1.05) | 1.06 (1.04 to 1.08) | 1.08 (1.06 to 1.10) | 1.15 (1.12 to 1.17) | <0.001 | 1.07 (1.06–1.08) |
| **CVD mortality** | | | | | | | |
| Number of deaths/participants | 8,455/104,224 | 6,683/104,224 | 6,994/104,224 | 7,492/104,224 | 9,123/104,224 | | |
| Person-years of follow-up | 1,467,932 | 1,481,515 | 1,468,962 | 1,463,029 | 1,425,659 | | |
| Model 1[a] | 1.00 | 0.85 (0.82 to 0.88) | 0.88 (0.85 to 0.91) | 0.93 (0.91 to 0.96) | 1.16 (1.12 to 1.19) | <0.001 | 1.15 (1.13–1.17) |
| Model 2[b] | 1.00 | 1.00 (0.97 to 1.03) | 1.01 (0.98 to 1.04) | 1.06 (1.03 to 1.10) | 1.15 (1.11 to 1.18) | <0.001 | 1.07 (1.06–1.09) |
| Model 3[c] | 1.00 | 1.02 (0.98 to 1.05) | 1.02 (0.99 to 1.06) | 1.09 (1.05 to 1.12) | 1.17 (1.13 to 1.21) | <0.001 | 1.08 (1.06–1.10) |
| Model 4[d] | 1.00 | 0.97 (0.93 to 1.01) | 0.96 (0.92 to 1.00) | 1.00 (0.95 to 1.04) | 1.01 (0.96 to 1.06) | 0.11 | 0.98 (0.95–1.00) |
| Model 5[e] | 1.00 | 1.00 (0.96 to 1.03) | 1.00 (0.97 to 1.04) | 1.06 (1.03 to 1.10) | 1.15 (1.11 to 1.19) | <0.001 | 1.08 (1.06–1.09) |
| Model 6[f] | 1.00 | 1.01 (0.97 to 1.04) | 1.01 (0.97 to 1.05) | 1.07 (1.03 to 1.11) | 1.14 (1.11 to 1.18) | <0.001 | 1.07 (1.06–1.09) |
| **Cancer mortality** | | | | | | | |
| Number of deaths/participants | 8,101/104,224 | 8,467/104,224 | 9,279/104,224 | 9,344/104,224 | 10,592/104,224 | | |
| Person-years of follow-up | 1,467,932 | 1,481,515 | 1,468,962 | 1,463,029 | 1,425,659 | | |
| Model 1[a] | 1.00 | 1.10 (1.06 to 1.13) | 1.20 (1.16 to 1.23) | 1.20 (1.16 to 1.24) | 1.39 (1.35 to 1.43) | <0.001 | 1.17 (1.15–1.18) |
| Model 2[b] | 1.00 | 1.07 (1.04 to 1.10) | 1.14 (1.10 to 1.17) | 1.13 (1.10 to 1.16) | 1.20 (1.16 to 1.24) | <0.001 | 1.09 (1.07–1.10) |
| Model 3[c] | 1.00 | 1.07 (1.04 to 1.11) | 1.14 (1.10 to 1.17) | 1.14 (1.10 to 1.18) | 1.21 (1.17 to 1.25) | <0.001 | 1.09 (1.07–1.10) |
| Model 4[d] | 1.00 | 1.00 (0.96 to 1.04) | 1.04 (1.00 to 1.08) | 1.02 (0.98 to 1.06) | 1.01 (0.96 to 1.06) | 0.94 | 0.98 (0.96–1.01) |
| Model 5[e] | 1.00 | 1.05 (1.01 to 1.08) | 1.10 (1.06 to 1.14) | 1.10 (1.07 to 1.14) | 1.15 (1.12 to 1.19) | <0.001 | 1.07 (1.05–1.08) |

(*Continued*)

**Table 2.** (Continued)

| Outcome and model | Quintiles of whole egg consumption | | | | | P trend | HR (95% CI) for half an egg/day |
|---|---|---|---|---|---|---|---|
| | Q1 | Q2 | Q3 | Q4 | Q5 | | |
| Model 6[f] | 1.00 | 1.06 (1.02 to 1.09) | 1.11 (1.08 to 1.15) | 1.11 (1.08 to 1.15) | 1.17 (1.13 to 1.21) | <0.001 | 1.07 (1.06–1.09) |

Values are HR (95% CI) unless stated otherwise.

[a]Model 1: adjusted for age and sex.

[b]Model 2: adjusted for model 1 + BMI (in kg/m$^2$; <18.5, 18.5 to 24.9, 25 to 29.9, 30 to 34.9, ≥35, or missing), race (white, black, Hispanic/Asian/Pacific Islander/American Indian/Alaskan native, or unknown/missing), education (less than high school, high school graduate, some college, college graduate, or unknown/missing), marital status (married/living as married or widowed/divorced/separated/never married/unknown), household income (quintiles), smoking (never smoked; quit, ≤20 cigarettes a day; quit, >20 cigarettes a day; currently smoking, ≤20 cigarettes a day; currently smoking, >20 cigarettes a day; or unknown), alcohol (0, 0.1–4.9, 5.0–29.9, ≥30 g/day), vigorous physical activity (never/rarely, 1–3 times/month, 1–2 times/week, 3–4 times/week, ≥5 times/week, or unknown/missing), usual activity at work (sit all day, sit much of the day/walk sometimes, stand/walk often/no lifting, lift/carry light loads, or carry heavy loads), and history of hypertension (yes or no), high cholesterol level (yes or no), heart disease (yes or no), stroke (yes or no), diabetes (yes or no), and cancer (yes or no) at baseline.

[c]Model 3: adjusted for model 2 + total energy and egg whites/substitutes.

[d]Model 4: adjusted for model 3 + dietary cholesterol.

[e]Model 5: adjusted for model 3 + red meat, fish, poultry, and dairy products, fruit, vegetables, potatoes, nuts/legumes, whole grains, refined grains, coffee, and sugar-sweetened beverages.

[f]Model 6: adjusted for model 3 + Healthy Eating Index–2015.

CI, confidence interval; CVD, cardiovascular disease; HR, hazard ratio; Q, quintile.

consumption and kidney disease mortality and between cholesterol intake and chronic liver disease mortality (both *P* for trend = 0.04). Egg white/substitute consumers had lower RD, AD, and chronic liver disease mortality compared with non-consumers (Table K in S1 Data).

## Substitution for whole eggs

In hypothetical substitution analyses, we found 6%, 8%, 9%, 7%, 13%, and 10% lower all-cause mortality when replacing half a whole egg (25 g/day) with equivalent amounts of egg whites/substitutes, poultry, fish, dairy products, nuts, and legumes, respectively (Fig 2). Replacing half a whole egg with an equivalent amount of egg whites/substitutes was associated with reductions of 3% in CVD mortality, 8% in cancer mortality, and 20% in RD mortality. Similarly, substituting an equivalent amount of nuts or legumes for half a whole egg was related to 9%–35% lower mortality from CVD, cancer, and RD. We also found significant changes in CVD, cancer, and RD mortality when replacing half a whole egg with an equivalent amount of poultry, fish, or dairy products.

## Subgroup analyses

In the secondary analysis of egg cooking method, we found similar positive associations with all-cause, CVD, cancer, and RD mortality for fried egg and non-fried egg consumption (Table L in S1 Data). In the subgroup analyses, although the associations of whole egg and cholesterol consumption with higher all-cause mortality persisted in all the subgroups, the positive associations were stronger among men, non-obese participants, never/former smokers, alcohol drinkers, non-hypertensive participants, non-diabetic participants, those with higher saturated fat intake, and those with longer duration of follow-up (Table 4).

## Sensitivity analyses

We observed similar associations when we further categorized the whole egg and cholesterol intakes by convenient consumption cutoffs (Tables M and N in S1 Data), excluded participants

**Table 3. Associations of dietary cholesterol intake with all-cause, CVD, and cancer mortality.**

| Outcome and model | Quintiles of dietary cholesterol intake | | | | | P trend | HR (95% CI) for 300 mg/ 2,000 kcal/day |
|---|---|---|---|---|---|---|---|
| | Q1 | Q2 | Q3 | Q4 | Q5 | | |
| Median intake (IQR) (mg/ 2,000/ kcal/day) | 118.3 (96.3 to 133.8) | 168.7 (158.2 to 178.8) | 207.9 (198.1 to 217.9) | 252.0 (239.7 to 265.9) | 330.0 (302.2 to 377.9) | | |
| **Total mortality** | | | | | | | |
| Number of deaths/participants | 23,431/104,224 | 23,935/104,224 | 24,847/104,224 | 26,341/104,224 | 30,774/104,224 | | |
| Person-years of follow-up | 1,482,386 | 1,477,276 | 1,468,371 | 1,456,999 | 1,422,065 | | |
| Model 1[a] | 1.00 | 1.04 (1.02 to 1.06) | 1.11 (1.09 to 1.13) | 1.21 (1.19 to 1.23) | 1.46 (1.43 to 1.48) | <0.001 | 1.53 (1.51–1.56) |
| Model 2[b] | 1.00 | 1.03 (1.01 to 1.05) | 1.08 (1.06 to 1.10) | 1.12 (1.10 to 1.14) | 1.22 (1.20 to 1.24) | <0.001 | 1.24 (1.22–1.26) |
| Model 3[c] | 1.00 | 1.04 (1.02 to 1.06) | 1.09 (1.07 to 1.12) | 1.15 (1.12 to 1.18) | 1.25 (1.21 to 1.29) | <0.001 | 1.26 (1.23–1.30) |
| Model 4[d] | 1.00 | 1.02 (1.00 to 1.04) | 1.06 (1.04 to 1.08) | 1.09 (1.07 to 1.12) | 1.17 (1.15 to 1.20) | <0.001 | 1.20 (1.17–1.22) |
| Model 5[e] | 1.00 | 1.02 (1.00 to 1.04) | 1.05 (1.03 to 1.07) | 1.09 (1.07 to 1.11) | 1.17 (1.14 to 1.19) | <0.001 | 1.19 (1.17–1.21) |
| Model 6[f] | 1.00 | 1.01 (0.99 to 1.03) | 1.03 (1.01 to 1.05) | 1.06 (1.04 to 1.09) | 1.14 (1.11 to 1.17) | <0.001 | 1.19 (1.16–1.22) |
| **CVD mortality** | | | | | | | |
| Number of deaths/participants | 7,206/104,224 | 7,223/104,224 | 7,218/104,224 | 7,872/104,224 | 9,228/104,224 | | |
| Person-years of follow-up | 1,482,386 | 1,477,276 | 1,468,371 | 1,456,999 | 1,422,065 | | |
| Model 1[a] | 1.00 | 1.02 (0.99 to 1.06) | 1.05 (1.01 to 1.08) | 1.17 (1.14 to 1.21) | 1.42 (1.37 to 1.46) | <0.001 | 1.50 (1.46–1.55) |
| Model 2[b] | 1.00 | 1.03 (1.00 to 1.07) | 1.05 (1.02 to 1.09) | 1.12 (1.09 to 1.16) | 1.21 (1.17 to 1.25) | <0.001 | 1.23 (1.19–1.26) |
| Model 3[c] | 1.00 | 1.03 (1.00 to 1.07) | 1.05 (1.01 to 1.09) | 1.11 (1.06 to 1.16) | 1.17 (1.10 to 1.24) | <0.001 | 1.16 (1.10–1.22) |
| Model 4[d] | 1.00 | 1.04 (1.00 to 1.07) | 1.05 (1.02 to 1.09) | 1.12 (1.08 to 1.16) | 1.19 (1.15 to 1.24) | <0.001 | 1.20 (1.16–1.24) |
| Model 5[e] | 1.00 | 1.02 (0.99 to 1.06) | 1.04 (1.00 to 1.07) | 1.10 (1.07 to 1.14) | 1.18 (1.14 to 1.22) | <0.001 | 1.20 (1.16–1.24) |
| Model 6[f] | 1.00 | 1.00 (0.97 to 1.04) | 1.01 (0.97 to 1.04) | 1.06 (1.02 to 1.10) | 1.12 (1.08 to 1.18) | <0.001 | 1.16 (1.11–1.21) |
| **Cancer mortality** | | | | | | | |
| Number of deaths/participants | 8,091/104,224 | 8,460/104,224 | 9,081/104,224 | 9,383/104,224 | 10,768/104,224 | | |
| Person-years of follow-up | 1,482,386 | 1,477,276 | 1,468,371 | 1,456,999 | 1,422,065 | | |
| Model 1[a] | 1.00 | 1.06 (1.03 to 1.10) | 1.17 (1.13 to 1.20) | 1.23 (1.20 to 1.27) | 1.46 (1.42 to 1.50) | <0.001 | 1.50 (1.47–1.54) |
| Model 2[b] | 1.00 | 1.04 (1.01 to 1.07) | 1.10 (1.07 to 1.14) | 1.12 (1.09 to 1.16) | 1.22 (1.19 to 1.26) | <0.001 | 1.25 (1.22–1.29) |
| Model 3[c] | 1.00 | 1.04 (1.01 to 1.08) | 1.12 (1.08 to 1.16) | 1.15 (1.10 to 1.21) | 1.28 (1.21 to 1.35) | <0.001 | 1.31 (1.25–1.37) |
| Model 4[d] | 1.00 | 1.02 (0.99 to 1.05) | 1.07 (1.04 to 1.10) | 1.08 (1.05 to 1.12) | 1.16 (1.12 to 1.20) | <0.001 | 1.19 (1.15–1.23) |
| Model 5[e] | 1.00 | 1.02 (0.99 to 1.06) | 1.08 (1.04 to 1.11) | 1.08 (1.05 to 1.12) | 1.16 (1.12 to 1.20) | <0.001 | 1.19 (1.15–1.23) |

(*Continued*)

**Table 3.** (Continued)

| Outcome and model | Quintiles of dietary cholesterol intake | | | | | *P* trend | HR (95% CI) for 300 mg/ 2,000 kcal/day |
|---|---|---|---|---|---|---|---|
| | Q1 | Q2 | Q3 | Q4 | Q5 | | |
| Model 6[f] | 1.00 | 1.03 (0.99 to 1.06) | 1.08 (1.05 to 1.12) | 1.09 (1.05 to 1.14) | 1.19 (1.14 to 1.24) | <0.001 | 1.24 (1.19–1.29) |

Values are HR (95% CI) unless stated otherwise.

[a]Model 1: adjusted for age and sex.

[b]Model 2: adjusted for model 1 + BMI (in kg/m$^2$; <18.5, 18.5 to 24.9, 25 to 29.9, 30 to 34.9, ≥35, or missing), race (white, black, Hispanic/Asian/Pacific Islander/ American Indian/Alaskan native, or unknown/missing), education (less than high school, high school graduate, some college, college graduate, or unknown/missing), marital status (married/living as married or widowed/divorced/separated/never married/unknown), household income (quintiles), smoking (never smoked; quit, ≤20 cigarettes a day; quit, >20 cigarettes a day; currently smoking, ≤20 cigarettes a day; currently smoking, >20 cigarettes a day; or unknown), alcohol (0, 0.1–4.9, 5.0–29.9, ≥30 g/day), vigorous physical activity (never/rarely, 1–3 times/month, 1–2 times/week, 3–4 times/week, ≥5 times/week, or unknown/missing), usual activity at work (sit all day, sit much of the day/walk sometimes, stand/walk often/no lifting, lift/carry light loads, or carry heavy loads), and history of hypertension (yes or no), high cholesterol level (yes or no), heart disease (yes or no), stroke (yes or no), diabetes (yes or no), and cancer (yes or no) at baseline.

[c]Model 3: adjusted for model 2 + total energy and intakes of cholesterol-containing foods (eggs, red meat, fish, poultry, and dairy products).

[d]Model 4: adjusted for model 2 + total energy and intakes of non-cholesterol-containing foods (fruit, vegetables, potatoes, nuts/legumes, whole grains, refined grains, coffee, and sugar-sweetened beverages).

[e]Model 5: adjusted for model 2 + total energy and Healthy Eating Index–2015.

[f]Model 6: adjusted for model 2 + total energy and intakes of saturated fat, polyunsaturated fat, monounsaturated fat, trans fat, animal protein, fiber, and sodium.

CI, confidence interval; CVD, cardiovascular disease; HR, hazard ratio, Q, quintile.

with extreme BMIs, or further adjusted for perceived health condition, a propensity score, the use of vitamin supplements and aspirin, or processed meat (Tables O–S in S1 Data). We also found largely unchanged associations when we further adjusted for the use of cholesterol-lowering medications; excluded the first 5 years of follow-up; excluded participants with CVD, cancer, or diabetes at baseline; or censored participants at 8 years of follow-up (Tables T–W in S1 Data).

## Discussion

We comprehensively examined the consumption of eggs and cholesterol in relation to mortality from different causes in 521,120 participants with a mean follow-up of 16 years. In this

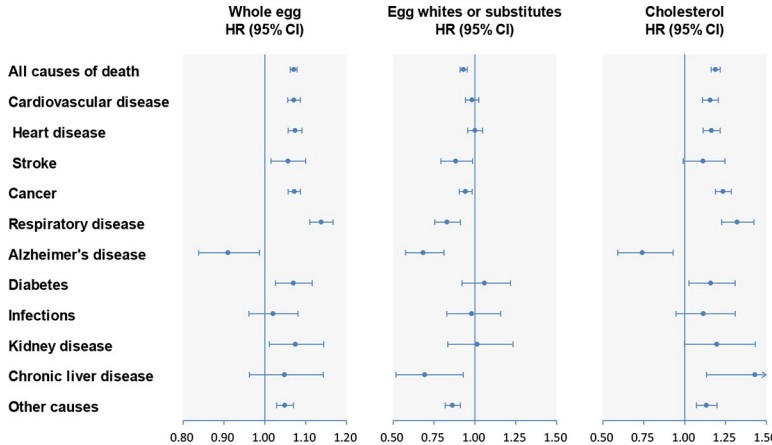

**Fig 1. Multivariable-adjusted HRs of all-cause and cause-specific mortality for whole egg, egg white/substitute, and cholesterol consumption.** Forest plots show the multivariable HRs of total and cause-specific mortality associated with each additional half a whole egg/day, egg white/substitute consumption (consumers versus non-consumers), or each additional 300 mg of cholesterol/day. HRs were adjusted for model 6 covariates for whole eggs, egg whites/ substitutes, and cholesterol. Horizontal lines represent 95% CIs. CI, confidence interval; HR, hazard ratio.

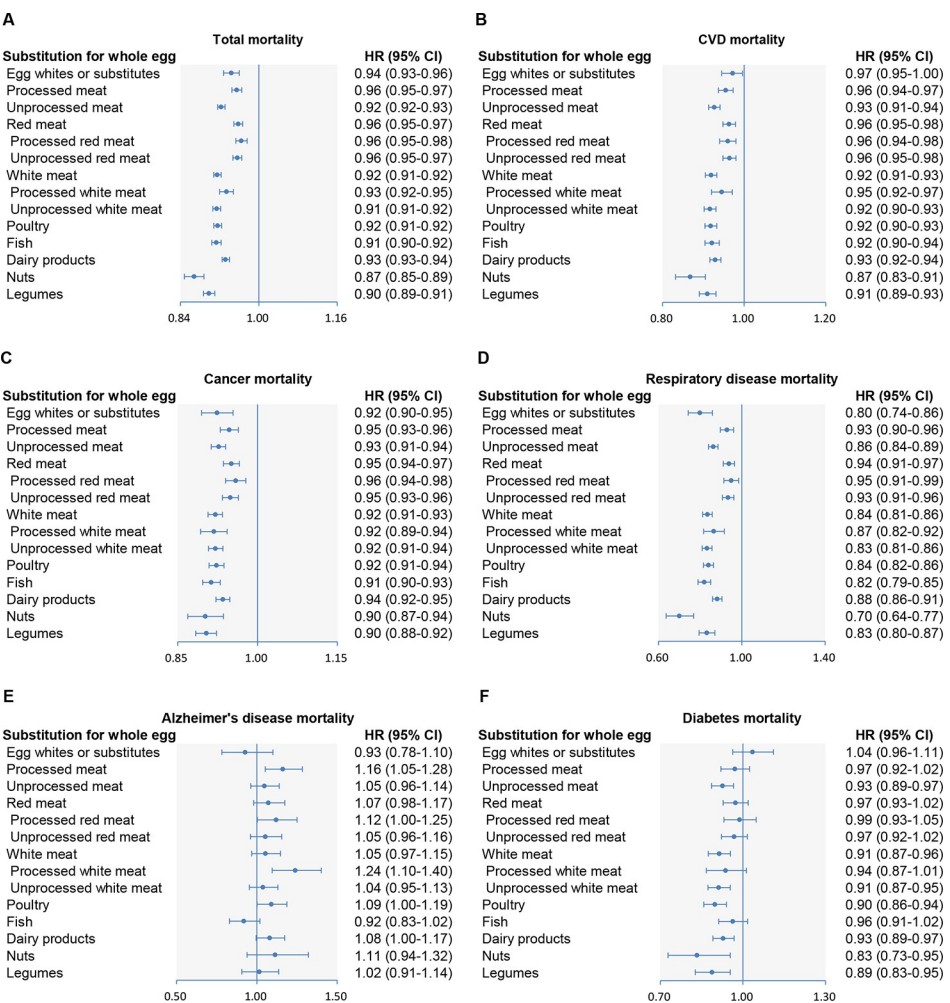

**Fig 2. Multivariable-adjusted hazard ratios of all-cause and cause-specific mortality by replacing 1 whole egg with equivalent amounts of other protein sources.** Forest plots show the multivariable HRs of (A) total, (B) CVD, (C) cancer, (D) respiratory disease, (E) Alzheimer disease, and (F) diabetes mortality associated with replacing half a whole egg/day with an equivalent amount of egg whites/substitutes, poultry, fish, dairy products, nuts, or legumes. HRs were adjusted for age; sex; BMI; race; education; marital status; household income; smoking; alcohol; vigorous physical activity; usual activity at work; history of hypertension, hypercholesteremia, heart disease, stroke, diabetes, and cancer at baseline; total energy intake; and intakes of fruit, vegetables, potatoes, whole grains, refined grains, coffee, and sugar-sweetened beverages. Horizontal lines represent 95% CIs. CI, confidence interval; CVD, cardiovascular disease; HR, hazard ratio.

large cohort, we showed that whole egg and cholesterol consumption were both associated with higher all-cause, CVD, and major cause-specific mortality, whereas egg white/substitute consumption was related to lower mortality. The increased all-cause, CVD, and cancer mortality associated with egg consumption was largely influenced by cholesterol intake. Egg white/substitute consumption was related to lower mortality. Replacing whole eggs with an equivalent amount of egg whites/substitutes or alternative protein sources was associated with lower overall mortality and mortality from CVD, cancer, and RD.

Evidence regarding egg consumption in relation to CVD or mortality has been limited and inconclusive [10–12]. Recent meta-analyses showed that frequent egg consumption (>7 eggs/week) compared to low consumption (<1 egg/week) was not significantly associated with higher all-cause mortality, due to seemingly contradictory outcomes among the included

**Table 4. Subgroup analyses for multivariable-adjusted HRs (95% CIs) of all-cause mortality.**

| | Baseline age | | | | |
| --- | --- | --- | --- | --- | --- |
| | ≥60 years | | <60 years | | P for interaction |
| | HR (95% CI) | P value | HR (95% CI) | P value | |
| Whole egg (25 g/2,000 kcal/day)[a] | 1.07 (1.06–1.08) | <0.001 | 1.09 (1.07–1.11) | <0.001 | 0.03 |
| Egg whites or substitutes (consumer versus non-consumer)[b] | 0.92 (0.90–0.95) | <0.001 | 0.96 (0.90–1.02) | 0.18 | 0.34 |
| Cholesterol (300 mg/2,000 kcal/day)[c] | 1.17 (1.14–1.20) | <0.001 | 1.27 (1.21–1.34) | <0.001 | 0.73 |
| | Sex | | | | |
| | Men | | Women | | P for interaction |
| | HR (95% CI) | P value | HR (95% CI) | P value | |
| Whole egg (25 g/2,000 kcal/day)[a] | 1.08 (1.07–1.09) | <0.001 | 1.05 (1.04–1.07) | <0.001 | 0.002 |
| Egg whites or substitutes (consumer versus non-consumer)[b] | 0.93 (0.90–0.96) | <0.001 | 0.93 (0.90–0.97) | 0.004 | 0.66 |
| Cholesterol (300 mg/2,000 kcal/day)[c] | 1.21 (1.17–1.24) | <0.001 | 1.14 (1.10–1.19) | <0.001 | 0.008 |
| | Baseline BMI | | | | |
| | ≥30 kg/m² | | <30 kg/m² | | P for interaction |
| | HR (95% CI) | P value | HR (95% CI) | P value | |
| Whole egg (25 g/2,000 kcal/day)[a] | 1.05 (1.04–1.07) | <0.001 | 1.08 (1.07–1.09) | <0.001 | <0.001 |
| Egg whites or substitutes (consumer versus non-consumer)[b] | 1.00 (0.95–1.05) | 0.99 | 0.91 (0.89–0.94) | <0.001 | <0.001 |
| Cholesterol (300 mg/2,000 kcal/day)[c] | 1.14 (1.09–1.18) | <0.001 | 1.22 (1.18–1.25) | <0.001 | <0.001 |
| | Current smoker | | | | |
| | Yes | | No | | P for interaction |
| | HR (95% CI) | P value | HR (95% CI) | P value | |
| Whole egg (25 g/2,000 kcal/day)[a] | 1.06 (1.04–1.07) | <0.001 | 1.08 (1.07–1.09) | <0.001 | <0.001 |
| Egg whites or substitutes (consumer versus non-consumer)[b] | 0.92 (0.86–0.99) | 0.02 | 0.93 (0.91–0.95) | <0.001 | 0.22 |
| Cholesterol (300 mg/2,000 kcal/day)[c] | 1.19 (1.13–1.24) | <0.001 | 1.19 (1.16–1.22) | <0.001 | <0.001 |
| | Alcohol drinker | | | | |
| | Yes | | No | | P for interaction |
| | HR (95% CI) | P value | HR (95% CI) | P value | |
| Whole egg (25 g/2,000 kcal/day)[a] | 1.08 (1.07–1.09) | <0.001 | 1.05 (1.04–1.07) | <0.001 | <0.001 |
| Egg whites or substitutes (consumer versus non-consumer)[b] | 0.92 (0.89–0.95) | <0.001 | 0.96 (0.92–1.00) | 0.03 | 0.03 |
| Cholesterol (300 mg/2,000 kcal/day)[c] | 1.22 (1.19–1.26) | <0.001 | 1.14 (1.10–1.18) | <0.001 | 0.003 |
| | Income level | | | | |
| | Below median | | At or above median | | P for interaction |
| | HR (95% CI) | P value | HR (95% CI) | P value | |
| Whole egg (25 g/2,000 kcal/day)[a] | 1.07 (1.06–1.08) | <0.001 | 1.07 (1.06–1.08) | <0.001 | 0.79 |
| Egg whites or substitutes (consumer versus non-consumer)[b] | 0.95 (0.92–0.98) | <0.001 | 0.91 (0.88–0.95) | <0.001 | 0.41 |
| Cholesterol (300 mg/2,000 kcal/day)[c] | 1.20 (1.16–1.23) | <0.001 | 1.17 (1.13–1.22) | <0.001 | 0.65 |
| | Fair/poor health | | | | |
| | Yes | | No | | P for interaction |
| | HR (95% CI) | P value | HR (95% CI) | P value | |
| Whole egg (25 g/2,000 kcal/day)[a] | 1.08 (1.06–1.09) | <0.001 | 1.07 (1.06–1.08) | <0.001 | 0.40 |
| Egg whites or substitutes (consumer versus non-consumer)[b] | 0.97 (0.93–1.02) | 0.27 | 0.92 (0.89–0.94) | <0.001 | 0.03 |
| Cholesterol (300 mg/2,000 kcal/day)[c] | 1.19 (1.14–1.24) | <0.001 | 1.18 (1.15–1.22) | <0.001 | 0.25 |
| | Hypertension | | | | |
| | Yes | | No | | P for interaction |
| | HR (95% CI) | P value | HR (95% CI) | P value | |
| Whole egg (25 g/2,000 kcal/day)[a] | 1.05 (1.04–1.07) | <0.001 | 1.07 (1.06–1.08) | <0.001 | 0.003 |
| Egg whites or substitutes (consumer versus non-consumer)[b] | 0.99 (0.94–1.03) | 0.50 | 0.94 (0.91–0.96) | <0.001 | 0.006 |
| Cholesterol (300 mg/2,000 kcal/day)[c] | 1.12 (1.07–1.17) | <0.001 | 1.19 (1.16–1.22) | <0.001 | <0.001 |

*(Continued)*

**Table 4.** (*Continued*)

|  | Baseline age | | | | |
| --- | --- | --- | --- | --- | --- |
|  | ≥60 years | | <60 years | | P for interaction |
|  | High cholesterol level | | | | |
|  | Yes | | No | | P for interaction |
|  | HR (95% CI) | P value | HR (95% CI) | P value | |
| Whole egg (25 g/2,000 kcal/day)[a] | 1.05 (1.03–1.07) | <0.001 | 1.07 (1.06–1.08) | <0.001 | 0.21 |
| Egg whites or substitutes (consumer versus non-consumer)[b] | 0.93 (0.88–0.99) | 0.02 | 0.96 (0.93–0.98) | <0.001 | 0.88 |
| Cholesterol (300 mg/2,000 kcal/day)[c] | 1.15 (1.10–1.21) | <0.001 | 1.18 (1.15–1.21) | <0.001 | 0.73 |
|  | Diabetes | | | | |
|  | Yes | | No | | P for interaction |
|  | HR (95% CI) | P value | HR (95% CI) | P value | |
| Whole egg (25 g/2,000 kcal/day)[a] | 1.07 (1.05–1.08) | <0.001 | 1.07 (1.06–1.08) | <0.001 | 0.01 |
| Egg whites or substitutes (consumer versus non-consumer)[b] | 1.00 (0.94–1.05) | 0.90 | 0.92 (0.89–0.94) | <0.001 | 0.002 |
| Cholesterol (300 mg/2,000 kcal/day)[c] | 1.18 (1.13–1.24) | <0.001 | 1.19 (1.16–1.22) | <0.001 | 0.01 |
|  | CVD | | | | |
|  | Yes | | No | | P for interaction |
|  | HR (95% CI) | P value | HR (95% CI) | P value | |
| Whole egg (25 g/2,000 kcal/day)[a] | 1.09 (1.07–1.11) | <0.001 | 1.07 (1.06–1.08) | <0.001 | 0.40 |
| Egg whites or substitutes (consumer versus non-consumer)[b] | 0.99 (0.95–1.03) | 0.54 | 0.90 (0.88–0.93) | <0.001 | 0.05 |
| Cholesterol (300 mg/2,000 kcal/day)[c] | 1.19 (1.14–1.25) | <0.001 | 1.19 (1.16–1.22) | <0.001 | 0.18 |
|  | Cancer | | | | |
|  | Yes | | No | | P for interaction |
|  | HR (95% CI) | P value | HR (95% CI) | P value | |
| Whole egg (25 g/2,000 kcal/day)[a] | 1.09 (1.07–1.11) | <0.001 | 1.07 (1.06–1.08) | <0.001 | 0.21 |
| Egg whites or substitutes (consumer versus non-consumer)[b] | 0.99 (0.93–1.06) | 0.82 | 0.92 (0.90–0.95) | <0.001 | 0.02 |
| Cholesterol (300 mg/2,000 kcal/day)[c] | 1.22 (1.15–1.30) | <0.001 | 1.18 (1.15–1.21) | <0.001 | 0.17 |
|  | HEI-2015 score | | | | |
|  | Below median | | At or above median | | P for interaction |
|  | HR (95% CI) | P value | HR (95% CI) | P value | |
| Whole egg (25 g/2,000 kcal/day)[a] | 1.07 (1.06–1.08) | <0.001 | 1.09 (1.07–1.10) | <0.001 | 0.002 |
| Egg whites or substitutes (consumer versus non-consumer)[b] | 0.92 (0.89–0.95) | <0.001 | 0.94 (0.91–0.97) | <0.001 | 0.59 |
| Cholesterol (300 mg/2,000 kcal/day)[c] | 1.18 (1.15–1.21) | <0.001 | 1.20 (1.15–1.25) | <0.001 | 0.25 |
|  | Saturated fat | | | | |
|  | Below median | | At or above median | | P for interaction |
|  | HR (95% CI) | P value | HR (95% CI) | P value | |
| Whole egg (25 g/2,000 kcal/day)[a] | 1.07 (1.06–1.09) | <0.001 | 1.06 (1.05–1.07) | <0.001 | 0.01 |
| Egg whites or substitutes (consumer versus non-consumer)[b] | 0.95 (0.93–0.98) | 0.002 | 0.97 (0.93–1.01) | 0.18 | <0.001 |
| Cholesterol (300 mg/2,000 kcal/day)[c] | 1.18 (1.13–1.24) | <0.001 | 1.16 (1.13–1.19) | <0.001 | 0.01 |
|  | Animal protein | | | | |
|  | Below median | | At or above median | | P for interaction |
|  | HR (95% CI) | P value | HR (95% CI) | P value | |
| Whole egg (25 g/2,000 kcal/day)[a] | 1.08 (1.06–1.09) | <0.001 | 1.06 (1.05–1.07) | <0.001 | 0.32 |
| Egg whites or substitutes (consumer versus non-consumer)[b] | 0.93 (0.90–0.96) | <0.001 | 0.97 (0.94–1.01) | 0.13 | <0.001 |
| Cholesterol (300 mg/2,000 kcal/day)[c] | 1.25 (1.20–1.30) | <0.001 | 1.13 (1.10–1.17) | <0.001 | 0.18 |
|  | Follow–up duration | | | | |
|  | ≥8 years | | <8 years | | P for interaction |
|  | HR (95% CI) | P value | HR (95% CI) | P value | |
| Whole egg (25 g/2,000 kcal/day)[a] | 1.06 (1.05–1.07) | <0.001 | 1.02 (1.01–1.04) | <0.001 | <0.001 |

(*Continued*)

**Table 4.** (Continued)

|  | Baseline age | | |  |
|---|---|---|---|---|
|  | ≥60 years | | <60 years | | P for interaction |
| Egg whites or substitutes (consumer versus non-consumer)[b] | 0.94 (0.92–0.97) | <0.001 | 1.01 (0.98–1.03) | 0.74 | <0.001 |
| Cholesterol (300 mg/2,000 kcal/day)[c] | 1.14 (1.11–1.17) | <0.001 | 1.06 (1.02–1.10) | 0.002 | <0.001 |

[a]HRs were adjusted for age; sex; BMI; race; education; marital status; household income; smoking, alcohol; vigorous physical activity; usual activity at work; history of hypertension, hypercholesteremia, heart disease, stroke, diabetes, and cancer at baseline; total energy; egg white/substitute consumption; and HEI-2015.

[b]HRs were adjusted for age; sex; BMI; race; education; marital status; household income; smoking; alcohol; vigorous physical activity; usual activity at work; history of hypertension; hypercholesteremia, heart disease, stroke, diabetes, and cancer at baseline; total energy; whole egg consumption; and HEI-2015.

[c]HRs were adjusted for age; sex; BMI; race; education; marital status; household income; smoking; alcohol; vigorous physical activity; usual activity at work; history of hypertension, hypercholesteremia, heart disease, stroke, diabetes, and cancer at baseline; total energy; and intakes of saturated fat, polyunsaturated fat, monounsaturated fat, trans fat, animal protein, fiber, and sodium.

CI, confidence interval; CVD, cardiovascular disease; HEI-2015, Healthy Eating Index–2015; HR, hazard ratio.

studies [13,14]. Similarly, another meta-analysis did not support a conclusive association between cholesterol intake and CVD outcomes [6]. However, the available studies in these meta-analyses had small sample sizes or few death cases and did not strengthen the methodological rigor to obtain precise effect estimates [6,13,14,29,30]. In addition, residual confounding could also be a reason for these inconsistent findings, given that egg consumption was commonly correlated with unhealthy lifestyle factors, including smoking, low physical activity, and unhealthy dietary patterns in Western countries [31], and that dietary cholesterol usually coexists with saturated fat and animal protein [2]. Our study carefully adjusted for major potential confounders and comprehensively assessed the significant associations among a large number of participants ($n$ = 521,120) and death cases ($n$ = 129,328), with a long-term duration of follow-up (16 years).

Our findings are consistent with a recent joint study of 6 prospective US cohorts ($n$ = 29,615) reporting that each additional half an egg/day was associated with 6%, 8%, and 8% higher risk of incident CVD, CVD mortality, and all-cause mortality, respectively [4]. Meanwhile, we reported a similar pattern for the associations for fried and non-fried egg consumption and found an inverse association of egg white/substitute consumption with mortality, underscoring the adverse role of egg yolk or cholesterol in premature deaths. These consistent results were further supported by our mediation model analyses, indicating that the associations of whole egg consumption with higher all-cause and CVD mortality were largely influenced by the cholesterol content. We also observed reductions in all-cause and major cause-specific mortality when replacing eggs with other alternative protein sources such as dairy, nuts, and legumes. Our results from the substitution analysis also suggested that meat containing less cholesterol was comparatively healthier than eggs, which is in line with the recent joint study, showing positive relationships between dietary cholesterol and all-cause and CVD mortality [4]. Correspondingly, avoiding 200 mg of dietary cholesterol per day decreased blood total cholesterol levels by 0.13 mmol/l and low-density lipoprotein cholesterol (LDL-c) levels by 0.10 mmol/l [32], although dietary cholesterol was not shown to significantly change serum triglyceride or very-low-density lipoprotein concentrations [6]. We observed that the positive associations were slightly weakened after adjusting for high blood cholesterol, which indicated high blood cholesterol might at least in part contribute to the adverse associations for egg/cholesterol. The latest meta-analysis summarizing results from 17 randomized controlled trials (RCTs) showed that egg consumption increased LDL-c and LDL-c/high-density lipoprotein cholesterol (HDL-c) ratio among healthy populations [33]. In addition,

evidence from Mendelian randomization studies and RCTs of LDL-lowering therapies consistently demonstrate that any mechanism of lowering LDL-c levels decreases the risk of atherosclerotic CVD, suggesting a causal role of LDL-c in atherosclerotic CVD development [34]. Together, accumulating evidence support the hypercholesterolemic effect (high LDL-c) as the biological mechanism for the effect of egg consumption and dietary cholesterol on CVD. The latest science advisory from the American Heart Association, developed after a review of human studies, highlighted that it remained advisable to limit cholesterol intake to current levels and that healthy individuals could include up to 1 whole egg per day [35]. Collectively, our findings strengthen the significance of minimizing dietary cholesterol intake among general populations for the management of cardiovascular health, and more clinical trials are warranted to collaborate our findings [36].

Regarding other cause-specific mortality, we found associations of whole egg and cholesterol consumption with higher cancer mortality. Dietary cholesterol has been reported to increase the incidence of breast and pancreatic cancers [37,38], while egg consumption is related to higher incidence of ovarian cancer [39]. For putative mechanisms, elevated total cholesterol and LDL-c levels have been linked with increased levels of pro-inflammatory cytokines, such as interleukin-6 and tumor necrosis factor-α [40]. In animal models, cholesterol may increase the generation of secondary bile acids and enhance cancer incidence [41,42]. Our finding of positive associations of whole egg and cholesterol intakes with diabetes mortality was consistent with a previous meta-analysis reporting higher risk of diabetes for those eating ≥3 eggs/week in US populations [43]. In addition, we detected adverse associations with risk of death from RD. A cross-sectional study reported that serum cholesterol level was positively associated with airway resistance in patients with chronic obstructive pulmonary disease [44]. In a murine model of asthma, dietary cholesterol enhanced pulmonary allergic inflammation [45]. Interestingly, we detected inverse associations of egg and cholesterol intakes with AD mortality, which was in accordance with the Spanish European Prospective Investigation into Cancer and Nutrition showing an inverse relationship between egg consumption and risk of death from nervous system causes [46]. Moderate egg consumption may have beneficial effects on certain areas of cognitive performance [47]. The protective effect on neurodegenerative diseases is potentially related to the content of dietary choline [48], zeaxanthin, and lutein [49], which may be beneficial to cognitive performance across the lifespan [50]. Excess cholesterol intake appears to be one of the main factors for the development of nonalcoholic fatty liver disease [51], which was supported by our current study showing a positive relationship between cholesterol intake and chronic liver disease mortality. Moreover, we detected associations of egg white/substitute consumption with lower stroke, cancer, RD, AD, and all-cause mortality, supporting that egg substitute intake instead of whole egg or egg yolk consumption should be encouraged for overall health [52].

Our subgroup analyses showed that the positive associations of whole egg and cholesterol consumption with mortality remained significant among all subgroups, including participants with a higher or lower quality diet, healthy participants, and those with a history of hypertension, high cholesterol level, CVD, cancer, or diabetes, although previous studies suggested the harmful effect of egg consumption might be restricted to diabetic patients [12,14]. The interaction between sex and egg/cholesterol intake on mortality could be due to sex dimorphisms in cholesterol catabolism and modulation [53]. Although alcohol drinkers are known to have higher HDL-c and lower LDL-c compared with non-drinkers [54], our results showed more pronounced positive associations of egg/cholesterol intake with mortality among alcohol drinkers. Smoking has been evidenced to exacerbate the effect of cholesterol on CVD risk [55]. However, we found slightly stronger positive associations among non-smokers. Future research is needed to verify and elucidate these observed interactions. We noted that a few

associations, such as for CVD and diabetes mortality, appeared stronger for non-fried eggs compared with fried eggs, which may be due to the type of fat used for frying [24]. For example, plant-sourced cooking oil consumption has been linked with lower risk of mortality in a nationwide cohort study [56].

The current investigation is to our knowledge the largest cohort study on this topic, with a large number of deaths from various causes and long-term follow-up with a high follow-up rate (>99%). In addition, when we excluded the first 5 years of follow-up or patients with major chronic diseases at baseline, the associations did not change materially, indicating the robustness of our results, with minimized reverse causality. Several limitations should also be noted. First, measurement errors of egg and cholesterol intakes assessed through postal surveys were inevitable despite our using a validated FFQ. However, such errors could have diluted real associations of egg and cholesterol intakes with mortality owing to the prospective study design. Second, egg and cholesterol intakes were assessed at baseline and could have potentially changed during the long-term follow-up. However, this might not appreciably affect our results because we detected similar associations when censoring participants at a shorter duration (8 years) of follow-up. Third, we could not specifically analyze the mediation of egg-sourced cholesterol for eggs due to the perfect correlation between egg-sourced cholesterol and eggs. Fourth, findings from this study might not be generalizable to non-US populations due to different nutrition and dietary patterns and different prevalence of chronic diseases. In addition, despite extensive adjustment for acknowledged diet and lifestyle risk factors, we cannot exclude the possibility of residual confounding from other unmeasured confounders. Lastly, a causal relationship cannot be established, given the study's observational nature.

## Conclusions

Among the US population, intakes of whole egg and cholesterol were associated with higher all-cause mortality and mortality from major causes including CVD, cancer, RD, and diabetes. Dietary cholesterol intake largely influenced all-cause, CVD, and cancer mortality associated with whole egg consumption. Replacing whole eggs with equivalent amounts of egg whites/substitutes or alternative protein sources was associated with lower all-cause, CVD, and major cause-specific mortality. Our findings support limiting cholesterol intake and suggest replacing whole eggs with egg whites/substitutes or other alternative protein sources for facilitating long-term survival and health. The 2015–2020 US dietary guidelines recommend eggs as part of a healthy diet without an upper limit on intake for cholesterol, which might lead to increases in cholesterol intake that could be detrimental to the prevention of premature death. Our results should be considered by clinicians and policy makers in updating dietary guidelines for Americans.

## Supporting information

**S1 STROBE Checklist. Strengthening the Reporting of Observational Studies in Epidemiology (STROBE) statement.**
(DOC)

**S1 Data. Supplementary data.**
(DOCX)

**S1 Text. Study protocol.**
(PDF)

## Acknowledgments

We are indebted to the participants in the NIH-AARP Diet and Health Study for their outstanding cooperation. We also thank Sigurd Hermansen and Kerry Grace Morrissey from Westat for study outcome ascertainment and management and Leslie Carroll at Information Management Services for data support and analysis. Cancer incidence data from the Atlanta metropolitan area were collected by the Georgia Center for Cancer Statistics, Department of Epidemiology, Rollins School of Public Health, Emory University, Atlanta, Georgia. Cancer incidence data from California were collected by the California Cancer Registry, Cancer Surveillance and Research Branch, California Department of Public Health, Sacramento, California. Cancer incidence data from the Detroit metropolitan area were collected by the Michigan Cancer Surveillance Program, Community Health Administration, Lansing, Michigan. The Florida cancer incidence data used in this report were collected by the Florida Cancer Data System, Miami, Florida, under contract with the Florida Department of Health, Tallahassee, Florida. The views expressed herein are solely those of the authors and do not necessarily reflect those of the Florida Cancer Data System or Florida Department of Health. Cancer incidence data from Louisiana were collected by the Louisiana Tumor Registry, Louisiana State University Health Sciences Center School of Public Health, New Orleans, Louisiana. Cancer incidence data from New Jersey were collected by the New Jersey State Cancer Registry, Rutgers Cancer Institute of New Jersey, New Brunswick, New Jersey. Cancer incidence data from North Carolina were collected by the North Carolina Central Cancer Registry, Raleigh, North Carolina. Cancer incidence data from Pennsylvania were supplied by the Division of Health Statistics and Research, Pennsylvania Department of Health, Harrisburg, Pennsylvania. The Pennsylvania Department of Health specifically disclaims responsibility for any analyses, interpretations, or conclusions. Cancer incidence data from Arizona were collected by the Arizona Cancer Registry, Division of Public Health Services, Arizona Department of Health Services, Phoenix, Arizona. Cancer incidence data from Texas were collected by the Texas Cancer Registry, Cancer Epidemiology and Surveillance Branch, Texas Department of State Health Services, Austin, Texas. Cancer incidence data from Nevada were collected by the Nevada Central Cancer Registry, Division of Public and Behavioral Health, Nevada Department of Health and Human Services, Carson City, Nevada.

## Author Contributions

**Conceptualization:** Jingjing Jiao, Yu Zhang.

**Data curation:** Jingjing Jiao, Yu Zhang.

**Formal analysis:** Pan Zhuang, Fei Wu, Lei Mao.

**Funding acquisition:** Jingjing Jiao, Yu Zhang.

**Investigation:** Pan Zhuang, Fei Wu, Lei Mao, Yiju Zhang, Xiaoqian Chen.

**Methodology:** Pan Zhuang, Jingjing Jiao, Yu Zhang.

**Software:** Pan Zhuang.

**Supervision:** Jingjing Jiao, Yu Zhang.

**Validation:** Fei Wu, Lei Mao, Yiju Zhang, Xiaoqian Chen.

**Writing – original draft:** Pan Zhuang, Jingjing Jiao, Yu Zhang.

**Writing – review & editing:** Pan Zhuang, Fei Wu, Lei Mao, Fanghuan Zhu, Yiju Zhang, Xiaoqian Chen, Jingjing Jiao, Yu Zhang.

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
