## [Editor Report · Decision Letter 0]

7 Sep 2020

Dear Dr Zhang, 

Thank you for submitting your manuscript entitled "Egg and Cholesterol Consumption and Mortality from Cardiovascular and Different Causes: Population-Based Prospective Cohort Study of 521,120 Participants" for consideration by PLOS Medicine.

Your manuscript has now been evaluated by the PLOS Medicine editorial staff and I am writing to let you know that we would like to send your submission out for external assessment.

Sincerely,

Richard Turner, PhD

Senior editor, PLOS Medicine

rturner@plos.org

---

## [Decision Letter · Decision Letter 1]

14 Oct 2020

Dear Dr. Zhang,

Thank you very much for submitting your manuscript "Egg and Cholesterol Consumption and Mortality from Cardiovascular and Different Causes: Population-Based Prospective Cohort Study of 521,120 Participants" (PMEDICINE-D-20-04351R1) for consideration at PLOS Medicine. 

Your paper was discussed among the editors and sent to independent reviewers, including a statistical reviewer. The reviews are appended at the bottom of this email and any accompanying reviewer attachments can be seen via the link below:

[LINK]

In light of these reviews, we will not be able to accept the manuscript for publication in the journal in its current form, but we would like to invite you to submit a revised version that fully addresses the reviewers' and editors' comments. You will appreciate that we cannot make a decision about publication until we have seen the revised manuscript and your response, and we expect to seek re-review by one or more of the reviewers. 

We hope to receive your revised manuscript by November 2. Please email us (plosmedicine@plos.org) if you have any questions or concerns.

Please let me know if you have any questions. Otherwise, we look forward to receiving your revised manuscript in due course. 

Sincerely,

Richard Turner, PhD

rturner@plos.org

Please revisit the data statement. It would appear that "No - there are some restrictions on access to data" would be appropriate, as you note restrictions in the subsequent section. 

Please remove the word "prospective" from the title (we think that the current study is a retrospective analysis of prospectively gathered data). You may wish to note that the analysis involves a US population. 

Please revisit the word "nationwide" at line 26. Around line 90 it appears that the cohort comprised people from 6 states and 2 cities, and we suggest briefly stating this information in the "methods and findings" section of your abstract instead. 

At line 28, we ask you to adopt the passive voice, i.e. "X participants were identified and prospectively followed ..." to avoid the implication that the present analysis is a prospective one.

At line 28 and elsewhere, please substitute "years".

We ask you to adapt the wording in your abstract to quote the dates of start and end of follow-up.

Please adapt the "Methods and findings" subsection of your abstract by adding a new final sentence, summarizing 2-3 of the study's main limitations. 

At line 48, please start the sentence with "In this study ..." or similar. 

After the abstract, we will need to ask you to add a new and accessible "author summary" section in non-identical prose. You may find it helpful to consult one or two recent research papers in PLOS Medicine to get a sense of the preferred style. 

Early in the methods section of your main text, please state whether the current analysis had a protocol or prespecified analysis plan, and if so attach the relevant document(s) as a supplementary file, referred to in the text. Please highlight analyses that were not prespecified. 

Please refer to the attached STROBE checklist early in your methods section ("See S1_STROBE_Checklist" or similar). 

At line 389, please remove the word "nationwide".

Please remove the information on funding, competing interests and data availability from the end of the main text. In the event of publication, this information will appear in the article metadata via entries in the submission form.

In your reference list, please ensure that 6 author names where appropriate are listed rather than 3, followed by "et al.".

In table 1, we suggest substituting "ethnicity" for "race".

Comments from the reviewers:

*** Reviewer #1: 

The paper by Zhuang and colleagues address an interesting research question, that is the relationship between egg intake and mortality, on which, to date, there is large inconsistency among studies. Moreover, the major novelty of the study is the mediation analysis which attempts to understand the nutrition-related mechanisms through which egg may exert their effect on health.

Overall the paper is of interest, however there are some suggestions that may improve the quality of the work. 

1. Eggs are natural sources not only of cholesterol, but also of saturated fat and protein. I would suggest the authors to test also these nutritional factors as potentially lining egg intake to mortality risk. 

2. The authors used total dietary cholesterol as a mediator of the egg-mortality relation. Is it correct? Shouldn't they see only the cholesterol contained in eggs rather than the whole cholesterol content of diet? How can this be disentangled?

3. The authors used the Mediate procedure in SAS. Why not relying on a counterfactual approach? Please, explain. 

4. In the mediation analysis, it is not clear to me if the proportion of effect explained by intermediate variables is obtained from a COX model including eggs as a categorical or continuous variable. This should be clarified. 

5. Also, were the HRs of the main analysis (Table 2) obtained from a COX PH or were those resulting from the mediate macro in SAS?

6. How many individuals are in the egg white /substitute consumers-non-consumers? It would be interesting to know. Moreover, egg white consumption may be a marker of socioeconomic status or healthy lifestyle, thus results may be affected by this bias. Please, provide more info on such type of consumers. 

7. The authors should acknowledge the study design (postal survey) as a major limitation of their work also highlighting main bias deriving from data collected through postal surveys. 

8. How do authors believe these data can be read and interpreted in light of national dietary guidelines that seem to recommend much higher intake of eggs per week? A discussion on this would be helpful from a public health perspective. 

9. Did the study collect blood samples? This would be interesting in light of analyzing which biological mechanisms underlie the association between egg and mortality. 

*** Reviewer #2: 

This is a very interesting and important study around the hot topic regarding the safety of egg and dietary cholesterol consumption. A large national sample was used and a comprehensive set of analyses was conducted. Compared with previous studies, the current one examined cooking methods, egg whites, and cause-specific mortality from non-CVD non-cancer causes. These are novel aspects. Findings are valuable to contribute to the discussion of health concerns of egg and dietary cholesterol intake, which has been debated intensely lately, particularly after the publication of the JAMA egg paper last year. However, there are a number of issues that need to be addressed to improve the clarity of the paper and to present clearer results. My comments are listed below for consideration.

1. Abstract: specify units are per 2000 kcal for estimates. In line 31, I assume there is only one FFQ used, so questionnaires should be questionnaire?

2. Lines 70-72, the interpretation of these guidelines is incorrect. It is not owing to the lack of evidence. There is evidence, but inconclusive or inconsistent. Please review the language in the 2013 AHA/ACC Guideline regarding dietary cholesterol. The background for the 2015-2020 Dietary Guidelines to remove the consumption limit for dietary cholesterol is because the mean consumption among US adults is already below 300 mg/day, so the focus can be placed on overall eating pattern instead of limiting cholesterol intake, although this recommendation is still debatable. Suggest revise the language. 

3. Lines 75-78: Unclear and confusing sentence, please revise. 

4. Lines 84-85: is quality of eggs a main exposure or amount of egg intake? What does quality of cholesterol intake mean? Did you study quality of cholesterol? This hypothesis does not fit in there at all. Suggest you delete it and replace it with the overall objective of your study. 

5. Line 110-112: The calibration methodology is unclear. Did the calibration help? What would be the associations if uncalibrated data were used?

6. Based on restricted cubic spline models in Fig B, the associations appeared perfectly linear. I suggest you could use continuous exposure variables and avoid using quintiles. The quintile approach has many statistical shortcomings and it uses a lot of space while presenting not much additional information. By using linear models and 25 grams of eggs and 300 mg of cholesterol per day, you will have space to include all primary exposures and mortality outcomes in the main manuscript. This is a major novelty of this study and such important results should not be hidden in the supplement. 

7. Many models are presented, but the primary model is not specified. To this reviewer, for eggs and egg whites, the primary model can be model 2 + total energy + all food groups. For dietary cholesterol, the primary model can be model 2 + total energy + HEI-2015. Other models are interesting, but these covariates are not confounders. For example, for dietary cholesterol model 3 which adjusted for all major cholesterol containing foods: dietary cholesterol comes from these foods. Adjusting for these foods left little variation for cholesterol. The cholesterol in the model represents cholesterol from unadjusted food sources such as egg-containing baked goods (e.g. cake). The authors need to carefully think about the meaning and interpretation of each model.

8. It is unclear if Spearman correlation analysis adjusted for energy.

9. Multiple comparisons are not adjusted. Consider to use <0.01 or <0.005 as the threshold to determine significance. 

10. Substitution analysis: Suggest separating unprocessed red meat and processed red meat. Combine processed red meat with processed poultry to create a processed meat group. Separate dairy products into low-fat and high-fat dairies if possible. Separate nuts and legumes. These food groups are very different, so combining them into border groups is not a good idea. 

11. The substitution unit is 50 grams per day. However, only about 2% of participants consumed this amount of eggs or more. For some foods such as nuts and legumes, their consumption amount may be similar to or lower than egg intake. Few people could meet the eligibility for substitution analysis. Suggest changing it to 25 grams/per day or around median egg intake in the sample. In addition, milk is possibly the largest food group among all dairy products. Do you consider people who do not eat eggs (solid food) likely choose to drink milk (fluid) instead?

12. Cooking method is understudied in the literature. It's interesting that results are largely similar between using fried and non-fried eggs. A few associations appeared stronger when using non-fried eggs. You may want to comment this finding in the discussion. 

13. Please remember to correct relevant places with copy-and-paste (e.g. Table B column head and Table E footnote Model 3). Please double check all tables and figures again. 

*** Reviewer #3: 

This is a large observational study of the association between egg/cholesterol consumption and cause-specific mortality in the US. The major strengths of the analysis are its large sample size, extensive dietary data and prolonged duration of follow-up, resulting in a large number of deaths for analysis. The study finds positive (and broadly similar) associations between both egg and cholesterol consumption and risk of all-cause, vascular and cancer mortality. The authors estimate that a substantial proportion of the associations with egg consumption are mediated by cholesterol and give hypthetical estimates of the effects of substituting eggs/cholesterol intake with other protein sources. Consumption of egg whites/substitutes was associated with lower mortality from several causes when compared with non use. This is a nice study. I have just a few comments the authors may wish to respond to in a revision.

1. Lack of specificity of the associations. One would probably have expected the associations with egg and cholesterol consumption to be stronger for vascular mortality than for cancer mortality, whereas for cholesterol the opposite is seen (ie, the association is strongest for cancer). How would the authors respond to the suggestion that this lack of specificity points towards residual confounding as a plausible explanation for at least some of the associations seen? Are there any causes of death the authors could use as 'negative controls' (ie, causes that are reliably NOT associated with egg/cholesterol consumption?) For example, external causes of death? Is this included in the 'other' category in Figure 1? If such a reliable negative control could be demonstrated then this would give additional reassurance to the validity of the main analyses. 

2. It seems that high blood cholesterol was consideed a confounder, but surely it is also (at least in part) a mediator of any association? Were the results materially different when analyses did NOT adjust for high cholesterol as a confounder?

3. Competing risks (ie, Fine-Gray) analyses were used as the primary method of analysis. Such methods estimate the sub-distribution hazard ratio and provide estimates of how cumulative incidence varies between the groups. They do not, however, estimate the cause-specific aetiological association (which is given by the simple Cox model that censors patients who die from an alternative cause at their date of death, ie, it assumes the hypothetical absence of death from other causes). Since the aetiolgical association is what is of most interest in this paper I would expect the primary analysis to be from a standard Cox model rather than a competing risk model. 

4. I was a bit surprised so see no discussion of how reported egg/cholesterol comsumption related to blood cholesterol, or to how blood cholesterol relates to vascular mortality risk (eg, by reference to the statin trials and/or Mendelian randomisation studies). Even if this is not possible from this study, could the authors summarise evidence on egg consumption and blood cholesterol?

MINOR COMMENTS

1. Line 84. I imagine the authors mean 'quantity' of eggs rather than 'quality'?

2. The end of the abstract suggests this study arises from within a trial but the paper describes a cohort study.

3. Line 182. I know what a covariance is but what is an invariance?

4. Table 2. The person-years is repeated in two separate rows. 

5. The appendix tables could be tidied up a bit with more thought given to how they are spaced.

*** Reviewer #4:

Zhuang et al have conducted a longitudinal study on egg and cholesterol consumption and total mortality and mortality from different causes within the frame of the NIH-AARP Diet and Health Study. The paper is well written, the analyses are thorough and the results are robust.

I have some comments which the authors may want to take into consideration.

Did the authors take egg consumption into account when they estimate the HEI-2015? It would be interesting to see the results without doing so.

Lines 167-168: They authors may want to impute missing data.

It would be interesting if the authors would present main results interim of servings instead of grams. 

As for the replacing methods, did they authors mean to replace one egg for one serving of the other foods/food groups? 

Also for the substitution analyses, it would be interesting to see poultry as replacement but without processed poultry products and also to see the substitution models using read meat and red meat products.

Have the authors considered using age as underlying time variable?

Did the authors have information on the type of fat that was used for frying the eggs? It may make a difference.

I think that the figure with the interaction analyses may be included in the main text.

As for the discussion, some further discussion on some causes of death such as respiratory diseases are needed since they may be less familiar to the readers.

Also, results for the interactions may be further discussed in the discussion.

Table 1: Please, consider removing the decimals in the dietary variables from animal protein to sugar-sweetened beverages. Also, please, include a measure of spread for the quantitative variables. 

Also in table 1, for the dietary variables, shouldn't the units be per day, e.g. fiber (g/day)?

Finally, I congratulate the authors for their manuscript.

***

[LINK]

---

## [Decision Letter · Decision Letter 2]

2 Dec 2020

Dear Dr. Zhang,

Thank you very much for re-submitting your manuscript "Egg and Cholesterol Consumption and Mortality from Cardiovascular and Different Causes: Population-Based Cohort Study of 521,120 Participants" (PMEDICINE-D-20-04351R2) for consideration at PLOS Medicine.

I have discussed the paper with editorial colleagues and our academic editor, and it was also seen again by 4 reviewers. I am pleased to tell you that, provided the remaining editorial and production issues are fully dealt with, we expect to be able to accept the paper for publication in the journal.

[LINK]

Please let me know if you have any questions. Otherwise, we look forward to receiving the revised manuscript shortly. 

Sincerely,

Richard Turner, PhD

rturner@plos.org

Requests from Editors:

Please remove the number of study participants from your title, and add "in the United States" or similar before the colon. 

At line 27, please make that "mortality from all causes ..." or similar. 

Please make that "an additional" at line 40.

Please substitute "years" for "y" in your abstract; and quote the mean or median age of participants, along with the proportion of female participants and summary information on ethnicity. 

Please restructure the end of the "Methods and Findings" subsection of your abstract so that the final sentence begins "Study limitations include ..." or similar, and that the limitations are summarized in a single sentence. 

At line 75, please make that "data on ..." or similar. 

At line 78, please make that "were positively associated ..."

At line 179, we note that you mention "later adjustments" to the analysis plan. We ask you to include some additional information here, explaining whether these were data-driven analyses. Please indicate in the Results section where findings are quoted for analyses that were not included in the prespecified analysis plan. 

In the Results section, please quote p values alongside HR and 95% CI, where available. 

At line 330, please make that "... were associated".

At line 352, please make that "Our findings are ...".

At line 427, please adapt the text to "... to our knowledge the largest".

Noting comments from one referee about the mediation analysis, we suggest retaining these analyses and briefly discussing the relevant limitations in the Discussion section of your main text. 

Please remove the information about funding from the end of the main text. This information will appear in the article metadata upon publication, via entries in the submission form. 

Throughout the text, please remove spaces from the reference call-outs (e.g., ... risk factors [23,24] ...).

Comments from Reviewers:

*** Reviewer #1: 

The authors have satisfactorily addressed my comments.

*** Reviewer #2: 

The authors have carefully revised their manuscript. I only have one minor comment: For Tables 2 and 3, please add an additional column showing results when eggs and dietary cholesterol are modeled as continuous variables to help compare results from different studies in the literature as well as to help partly address the issue that the quintile cutoffs from different studies may vary considerably. 

Victor Wenze Zhong, Ph.D.

Assistant Professor

Division of Nutritional Sciences

Cornell University

*** Reviewer #3: 

Thanks. No further comments.

*** Reviewer #4: 

Thank you for your thorough work in addressing the previously raised comments.

My main concern with the current version of the manuscript has to do with the mediation analysis. So, it seems that egg consumption may lead to increased total dietary cholesterol intake (not just from eggs) and thus lead to an increased risk in the ascertained outcome. I believe this might not be completely accurate and I would suggest not to include these analyses since they might be misleading or at least consider them as ancillary.

***

[LINK]

---

## [Editor Report · Decision Letter 3]

5 Jan 2021

Dear Dr. Zhang,

I am writing concerning your manuscript submitted to PLOS Medicine, entitled “Egg and Cholesterol Consumption and Mortality from Cardiovascular and Different Causes in the United States: A Population-Based Cohort Study of 521,120 Participants.”

We have now completed our final technical checks and have approved your submission for publication. You will shortly receive a letter of formal acceptance from the editor.

Kind regards,

PLOS Medicine